# Adsorption, Antibacterial and Antioxidant Properties of Tannic Acid on Silk Fiber

**DOI:** 10.3390/polym11060970

**Published:** 2019-06-03

**Authors:** Wen Zhang, Zhi-Yi Yang, Xian-Wei Cheng, Ren-Cheng Tang, Yi-Fan Qiao

**Affiliations:** 1National Engineering Laboratory for Modern Silk, College of Textile and Clothing Engineering, Soochow University, 199 Renai Road, Suzhou 215123, China; wzhang0219@stu.suda.edu.cn (W.Z.); chengxian-wei@outlook.com (X.-W.C.); 18340872728@163.com (Y.-F.Q.); 2Lushan College, Guangxi University of Science and Technology, Liuzhou 545000, China; yzy166@yahoo.com.cn

**Keywords:** tannic acid, silk, antibacterial activity, antioxidant activity, adsorption

## Abstract

Natural bioactive compounds have received increasing attention in the functional modification of textiles. In this work, tannic acid was used to impart antibacterial and antioxidant functions to silk using an adsorption technique, and the adsorption properties of tannic acid on silk were studied. The adsorption quantity of tannic acid on silk increased with decreasing pH in the range of 3–7. The rates of the uptake of tannic acid by silk were well correlated to the pseudo-second-order kinetic model, and the calculated activation energy of adsorption was 93.49 kJ/mol. The equilibrium adsorption isotherms followed the Langmuir model. The adsorption rate and isotherm studies demonstrated that the chemical adsorption of tannic acid on silk occurred through the ion-ion interaction between tannic acid and silk. Tannic acid displayed good building-up properties on silk. The silk fabric treated with 0.5% tannic acid (relative to fabric weight) exhibited excellent and durable antibacterial properties. Moreover, the silk fabrics treated with 2% and 5% tannic acid had good and durable antioxidant properties. The treatment by tannic acid had less impact on the whiteness of the silk fabric. In summary, tannic acid can be used as a functional agent for preparing healthy and hygienic silk materials.

## 1. Introduction

People have come to realize the important role of environmental protection owing to the aggravation of environmental destruction. In order to get the resources and environments onto the right track of circulation, the substitution for non-regenerated resources with regenerated resources has drawn increasing interest in the textile industry [1,2]. Bioactive products derived from natural resources, for instance flavonoids [3], chitosan [4], oxidized cellulose and carboxycellulose [5,6], and vitamins [7], are gaining great attention owing to their non-toxic, degradable, and eco-friendly performance. Many studies have also proven the promising application prospects of natural bioactive products in promoting the antibacterial, antioxidant, and anti-inflammatory activities of hygiene-related and medical textiles [2,8,9,10].

Silk is one of the earliest natural protein fibers used by humans, known as the “queen of fibers” because of its luxury appearance and excellent wearing comfort. The good biocompatibility, non-toxicity, and superior mechanical property of silk make it attract more and more attention in the field of biomedical materials and biological textiles [11,12]. Nevertheless, the poor antibacterial and antioxidant properties of silk severely constrain its practical application in daily clothing as well as hygienic and medical materials. Silk provides good environments for the growth of micro-organisms. The microbial growth and multiplication cause the generation of fetid odors and mildew, the deterioration and discoloration of silk fiber, skin infection, and allergic responses [2]. A series of synthetic and natural antibacterial agents have been applied in the antibacterial treatment of silk, including quaternary ammonium compounds [13], inorganic nano-materials [14], chitosan [15,16], and natural bioactive compounds [3,17]. On the other hand, the active ingredients of natural plant extracts (e.g., curcumin, baicalin, quercetin, rutin, and chlorogenic acid) have been used to upgrade the antioxidant property of silk [3,8,18,19]. The silk fabrics and materials containing natural antioxidant compounds act as a reservoir system progressively delivering the active substances to the skin layers when in use [20]. The active substances released from silk can deactivate highly reactive and harmful species (e.g., active oxygen radicals) caused by skin degeneration and present in the atmosphere, and thus protect the skin from oxidative stress, inflammation, and aging [21,22]. Silk is frequently in direct contact with skin during its use, and thus the introduction of antibacterial and antioxidant functions is of great significance for developing healthy and hygienic silk textiles and materials.

Tannins are non-toxic, eco-friendly and polyphenolic phytochemicals that are widely found in beverages and foodstuffs, particularly in red wine, green tea, raspberries, and black-eyed peas [23,24,25]. Tannic acid is one of the hydrolysable tannins, which is composed of a central glucose molecule and 10 galloyl groups, and its chemical structure is shown in Figure 1 [23]. Previous literature has reported the application of tannic acid in the dyeing, pickling, and tanning processes for leather products [26]. Recently, in food and medical sciences, tannic acid has been confirmed to be efficient in deactivating free radicals and defending against microorganisms [27,28]. Besides, due to its extremely pale color and multi-functional groups, tannic acid is very appropriate for the functional modification of textile materials. The complex of tannic acid, metal ions, and cellulose was used for improving the antibacterial property of jute fabric [29]. However, very few studies clearly described the mechanism of the adsorption of tannic acid by textile fibers and the functionalization of silk fabric by tannic acid.

The aim of this article was to prepare bioactive and durable silk materials by the introduction of tannic acid using a facile adsorption process. In this work, firstly, the effects of pH and temperature on the uptake of tannic acid by silk were explored. Subsequently, the adsorption process and mechanism of tannic acid were studied using kinetic equations and thermodynamic models to fit the adsorption data. The building-up property of tannic acid was also determined with the aim of understanding its utilization rate. Additionally, the whiteness and yellowness of the treated silk fabric were measured to investigate the influence of tannic acid on the color of silk. Finally, the antibacterial and antioxidant activities, as well as the washing durability of the treated silk fabric, were analyzed.

## 2. Materials and Methods

### 2.1. Materials

The scoured silk fabric of crepe de Chine with the following specifications was bought from Suzhou Jiaduoli Silk Apparel Co. Ltd., Suzhou, China: Warp and weft count, 23.3 dtex/2; warp density, 42 threads/cm; weft density, 60 threads/cm; and weight per unit area, 52 g/m^2^. Tannic acid was purchased from Aladdin Industrial Corporation Co. Ltd., Shanghai, China. 2,2′-Azinobis (3-ethylbenzothiazoline-6-sulfonic acid) diammonium salt (ABTS) was bought from Sigma–Aldrich Trading Co. Ltd., Shanghai, China. Citric acid, disodium hydrogen phosphate, sodium dihydrogen phosphate, potassium dihydrogen phosphate, potassium persulfate, and ethanol were of analytical reagent grade. Nutrient agar and nutrient broth were bought from Sinopharm Chemical Reagent Co. Ltd., Shanghai, China, and Shanghai Sincere Biotech Co. Ltd., Shanghai, China, respectively. A commercial detergent for silk was bought from Shanghai Zhengzhang Laundering and Dyeing Co. Ltd., Shanghai, China.

### 2.2. Adsorption Experiments

All the adsorption of tannic acid was carried out in a XW-ZDR low-noise oscillated dyeing machine (Jingjiang Xinwang Dyeing and Finishing Machinery Factory, Jingjiang, China) equipped with time and temperature controllers. The liquor ratio (the ratio of liquor volume to fabric weight) was kept at 50:1. The pH of the tannic acid solution was adjusted by McIlvaine buffer (citric acid and disodium hydrogen phosphate mixture) and detected by the PHS-3C pH meter (Shanghai REX Instrument Factory, Shanghai, China).

#### 2.2.1. Effect of pH and Temperature on the Adsorption of Tannic Acid

To examine the effect of pH on the adsorption of tannic acid, silk fabrics were treated with 10% owf (on the weight of fabric) tannic acid in the pH range of 3–7, and the temperature was raised to 90 °C from 25 °C at a rate of 2 °C/min, and at 90 °C the treatment continued for 60 min. To examine the effect of temperature on the adsorption of tannic acid, silk fabrics were immersed with 10% owf tannic acid at pH 3, and the temperature was raised from 50 to 90 °C at a rate of 2 °C/min and then held for 60 min.

#### 2.2.2. Adsorption Kinetics and Adsorption Isotherms of Tannic Acid

For the adsorption kinetics of tannic acid, silk fabrics were treated with a 10% owf tannic acid solution at pH 3 at various constant temperatures (70, 80, and 90 °C) for different times (1 to 120 min). For the experiments of the equilibrium adsorption isotherms of tannic acid, silk fabrics were treated with various initial concentrations (1% to 30% owf) of tannic acid solutions at pH 3 and constant temperatures (70, 80, and 90 °C) for 120 min.

#### 2.2.3. Building-Up Property of Tannic Acid

The building-up property of tannic acid on silk was measured using a series of tannic acid concentrations (1–25%) at pH 3. The temperature was raised from 25 °C to 90 °C at a rate of 2 °C/min, and at 90 °C the treatment continued for 60 min. The as-prepared samples were also employed for the evaluation of antioxidant and antibacterial activities.

### 2.3. Measurements

#### 2.3.1. Uptake of Tannic Acid on Silk

The absorbance of tannic acid solutions was measured with the Shimadzu UV-1800 UV-Vis spectrophotometer (Shimadzu Co. Ltd., Kyoto, Japan). The concentration of tannic acid was determined by reference to its calibration curve at the maximum adsorption wavelength of 276 nm. The determination of the exhaustion of tannic acid was based on Equation (1),
(1)$$\mathrm{Exhaustion}\;(\%)=100\times \frac{m_{0}-m_{1}}{m_{0}},$$
where *m*_0_ and *m*_1_ are the quantities of tannic acid in the solution before and after treatment, respectively. The quantity of tannic acid on silk was calculated by taking into account the initial and final concentrations of tannic acid in the solution as well as the weight of the dried silk fabric.

#### 2.3.2. Whiteness and Yellowness Indexes

The color space coordinates (lightness (*L*), redness–greenness (*a*), blueness–yellowness (*b*), chroma (*C*)), and tristimulus values (*X*, *Y*, and *Z*) of the untreated and treated silk fabrics were measured by the HunterLab UltraScan PRO reflectance spectrophotometer (Hunter Associates Laboratory Inc., Reston, VA, USA) using the D65 illuminant and 10° standard observer. Each sample was folded two times so as to get four layers, and the average of five measurements was recorded. The Hunter whiteness index and yellowness index were calculated according to Equations (2) and (3), respectively:(2)$$\mathrm{Whiteness}\;\mathrm{index}\;=\;100-{\left[{\left(100-L\right)}^{2}+a^{2}+b^{2}\right]}^{{}^{0.5}};\;\mathrm{and}$$
(3)Yellowness index= 100×(1−0.847Z/Y).

#### 2.3.3. Antioxidant Activity

The antioxidant activity of treated fabrics was evaluated by the ABTS radical decolorization assay as per the previously reported method [8,30]. A solution of ABTS radical cation (ABTS˙^+^) was prepared by the reaction between ABTS (7 mM) stock solution and potassium persulfate (2.45 mM), and was stored in the dark for 12 to 16 h at room temperature. Later, the ABTS˙^+^ regent was diluted with a phosphate buffer (0.1 M, pH 7.4) to reach an absorbance of 0.700 ± 0.025 at 734 nm. For the spectrophotometric measurement of the samples, 10 mL of ABTS˙^+^ solution and 10 mg of fabric sample were mixed. After 30 min, the scavenging capability of ABTS˙^+^ at 734 nm was calculated using Equation (4),
(4)$$\mathrm{Antioxidant}\;\mathrm{activity}\;(\%)\;=\;100\times \frac{A_{ctrl}-A_{spl}}{A_{ctrl}},$$
where *A_ctrl_* is the initial absorbance of the ABTS˙^+^ and *A_spl_* is the absorbance of the remaining ABTS˙^+^ in the presence of the fabric sample. The average antioxidant activity of three parallel samples was used.

#### 2.3.4. Antibacterial Activity

*Escherichia coli* (*E*. *coli*, Gram negative) and *Staphylococcus aureus* (*S*. *aureus*, Gram positive) were employed to evaluate the antibacterial activity of samples according to GB/T 20944.3–2008 [8,31]. The fragments of treated fabrics (0.75 g) were added into the conical flasks with bacteria and placed in the incubator for 24 h. The incubator temperatures were 30 °C for the *E. coli* solution and 24 °C for the *S. aureus* solution, respectively. Later, the bacteria solutions were diluted a 1000 times with a sterilizing phosphoric buffer solution so as to prepare a test bacteria solution. The above bacteria solutions were inoculated into petri dishes with nutrient agar and cultured at 37 °C for 24 h for the *E. coli* solution, and for 48 h for the *S. aureus* solution. Finally, the bacterial colonies growing on the petri dishes were recorded, and the antibacterial activity was calculated according to Equation (5),
(5)$$\mathrm{Antibacterial}\;\mathrm{activity}\;(\%)\;=\;100\times \frac{N_{ctrl}-N_{spl}}{N_{ctrl}},$$
where *N_ctrl_* and *N_spl_* are the quantities of the visual bacterial colonies of the standard cotton fabric and the tested fabric, respectively. The average antibacterial activity of two parallel samples was used.

#### 2.3.5. Washing Durability

The washing durability of the treated fabric was evaluated by determining the antioxidant and antibacterial activities of samples subjected to different washing cycles. The washing test was carried out in the pots housed in a Wash Tec-P Fastness Tester (Roaches International, West Yorkshire, UK). The washing solution contained 2 g/L commercial detergent, and the liquor ratio was 50:1. Each washing was conducted at 40 °C for 30 min. After one washing, the fabric was removed, gently squeezed, and rinsed with tap water.

## 3. Results and Discussion

### 3.1. Adsorption Properties of Tannic Acid on Silk

#### 3.1.1. pH and Temperature Dependence of Tannic Acid Adsorption

The pH value of a bath can affect the dissociation of phenolic hydroxyl groups in tannic acid, the ionization of amino group in silk fiber, and the hydrolysis of silk fiber, thus exerting an impact on the adsorption of tannic acid and the tensile strength of silk fabric. In our previous researches [3,8], the tensile strength loss of silk fabric was found when it was treated with natural flavonoids at pH 2.75 and at 90 °C. Taking this fact into consideration, low pH was not selected in the present research. The influences of pH and temperature on tannic acid adsorption are depicted in Figure 2. As shown in Figure 2a, the adsorption capacity of tannic acid on silk depended appreciably on the pH of the tannic acid solution. The uptake of tannic acid by silk gradually increased with decreasing pH; the exhaustion percentage increased to 80.5% at pH 3 from 37.4% at pH 7. This implies that tannic acid can be adsorbed by silk by virtue of the ion-ion interaction between the positively charged amino groups in silk and the negatively charged hydroxyl groups in tannic acid.

Figure 2b shows the exhaustion of tannic acid between 50 and 90 °C. With the temperature rising from 50 to 70 °C, the adsorption of tannic acid on silk fiber showed an upward trend, which is attributed to the increase in the swelling degree of silk fiber and the kinetics energy of tannic acid molecules, leading to the increased capability of the diffusion of tannic acid into the interior of the silk fiber. When the temperature increased from 70 to 90 °C, the adsorption of tannic acid showed a downward trend. This phenomenon is related to the decreasing affinity of tannic acid to silk fiber with increasing temperature.

#### 3.1.2. Adsorption Kinetics of Tannic Acid

The adsorption process of tannic acid on silk fiber was investigated by virtue of the function of the adsorption amount (*C_t_*) of tannic acid against time (*t*). Figure 3a shows the adsorption rate curves of tannic acid on silk fiber at 70, 80, and 90 °C. The amount of tannic acid grew sharply in the first 20 min, and remained steady after 60 min, implying that the adsorption process reached equilibrium. Furthermore, the growth of temperature induced an increase in the adsorption rate of tannic acid in the initial adsorption process, but reduced the equilibrium adsorption quantity of tannic acid.

To further study the mechanism of the adsorption process of tannic acid on silk fiber, the pseudo-second-order kinetic equation (Equation (6)) [32] was used to simulate the experiment data at different temperatures:(6)$$\frac{t}{C_{t}}=\frac{1}{kC_{\infty }^{2}}+\frac{1}{C_{\infty }}t,$$
where *k* (g/mg/min) is the rate constant of adsorption, and *C*_∞_ (mg/g) and *C_t_* (mg/g) are the quantity of adsorbed tannic acid on silk at equilibrium and at time *t*, respectively.

The pseudo-second-order kinetic model is a common method used to represent the chemisorption involving valency forces through the sharing or exchange of electrons between the adsorbent and adsorbate as a covalent force and ion exchange [32]. Figure 3b shows the plots of *t*/*C_t_* vs. *t*, suggesting the linear correlation of data. The slope and intercept of the linear regression line were used to calculate the *k* and *C*_∞_. Additionally, the half-adsorption time (*t*_1/2_) and initial adsorption rate (*r*_i_) were given by Equations (7) and (8), respectively:(7)$$t_{1/2}=\frac{1}{kC_{\infty }};\;\mathrm{and}$$
(8)$$r_{i}=kC_{\infty }{}^{2}.$$

The adsorption kinetic parameters obtained from the pseudo-second-order model are listed in Table 1. It is clear that correlation coefficient (*R*^2^) values of the linear plots were higher than 0.999, indicating that the adsorption of tannic acid on silk fiber complies with the pseudo-second-order kinetic equation. In other words, the adsorption of tannic acid on silk fiber belongs to chemical adsorption. At higher temperatures, tannic acid exhibited a higher initial-adsorption rate, a shorter half-adsorption time, and a higher equilibrium rate constant which are associated with the higher extent of fiber swelling and the higher diffusion energy of tannic acid molecules. However, the equilibrium adsorption quantity of tannic acid showed a significant decrease from 91.74 mg/g at 70 °C to 76.92 mg/g at 90 °C. This can be explained by the decreasing affinity of tannic acid to silk fiber with increasing temperature.

The adsorption rate constants (*k*) at different temperatures in Table 1 were employed to calculate the apparent activation energy of the adsorption of tannic acid using the Arrhenius equation (Equation (9)) [33]:(9)$$\ln\;k=\;\ln\;A-\frac{E_{a}}{RT},$$
where *E*_a_ (kJ/mol), *A*, *R*, and *T* (K) refer to the Arrhenius activation energy, Arrhenius factor, gas constant, and absolute temperature, respectively.

As shown in Figure 4, the *R*^2^ value for the plot of ln*k* vs. 1/*T* was 0.9994, proving the validity of the Arrhenius equation in estimating the activation energy of the adsorption of tannic acid. The slope of the plot was used to determine the activation energy *E*_a_, giving a result of 93.49 kJ/mol. In general, the activation energy of chemical adsorption is between 40 and 800 kJ/mol [34,35]. This reveals that the adsorption of tannic acid on silk fiber is a chemical adsorption, in accordance with the result obtained from the pseudo-second-order kinetic model.

#### 3.1.3. Equilibrium Adsorption Isotherms of Tannic Acid

The adsorption isotherms of tannic acid on silk fiber at pH 3 are depicted in Figure 5a, showing a decreasing trend in equilibrium adsorption capacity with the growth of temperature. In other words, high temperature is not conducive to the adsorption of tannic acid on silk fiber. The equilibrium adsorption isotherms are the basis for understanding the interactions between adsorbent and adsorbate, and they also provide the most important thermodynamics information for the design and analysis of adsorption process. Herein, the Langmuir and Freundlich isotherm models were employed to fit the experimental adsorption data.

The Langmuir isotherm describes the complete mono-layer adsorption of adsorbate onto a series of equivalent sites located on the surface of the adsorbent, which is one of the most common chemisorption processes. The Langmuir adsorption can be expressed using Equation (10),
(10)$$C_{f}=\frac{SK_{L}C_{s}}{1+K_{L}C_{s}},$$
where *C*_f_ (mg/g) and *C*_s_ (mg/L) represent the concentration of tannic acid on silk and in solution at equilibrium, respectively, *S* is the saturation concentration of tannic acid on silk, and *K*_L_ is the Langmuir affinity constant.

The Freundlich isotherm is a widely accepted multi-site adsorption isotherm for heterogeneous surface system. The expression of Freundlich adsorption is shown in Equation (11),
(11)$$C_{f}=K_{F}C_{s}^{n},$$
where *K*_F_ is the Freundlich affinity constant and *n* is an indicator of surface heterogeneity or adsorption intensity.

The Langmuir and Freundlich adsorption parameters calculated by the linear plots of 1/*C*_f_ vs. 1/*C*_s_ (Figure 5b) and ln*C*_f_ vs. ln*C*_s_ (not shown), respectively, are listed in Table 2. As shown in Table 2, the correlation coefficients (*R*^2^) of Langmuir model were higher than 0.989, whereas the Freundlich model displayed low correlation coefficients, suggesting that the Langmuir model is more favorable than the Freundlich model. Previous studies also found that the Langmuir model fitted well with the adsorption of tannic acid on positively charged amino-functionalized magnetic nanoadsorbent and cetylpyridinium bromide-modified zeolites [36,37]. Therefore, it seems to be reasonable to conclude that the adsorption of tannic acid on silk occurs by means of the ion-ion interactions between the deprotonated amino groups in silk fiber and the ionized phenolic hydroxyl groups in tannic acid molecules under acid conditions, and follows the ideal monolayer chemisorption.

The dissociation equilibrium constant (p*K*_a_) of tannic acid is 7–8 (at this pH, the ionized fraction of phenolic hydroxyl groups corresponds to 50%) [38]. In the present research, the adsorption isotherm experiments were carried out at pH 3. At this pH, the partially ionized phenolic hydroxyl groups in tannic acid can be adsorbed by the positively charged amino groups in silk fiber by virtue of electrostatic interactions. It can be supposed that the adsorption of the ionized tannic acid by silk fiber breaks the ionization equilibrium of tannic acid in solution, and consequently promotes the further ionization of tannic acid and the continuous adsorption of tannic acid on silk fiber.

Table 2 shows that the *K*_L_ and the *S* values decreased with increasing temperature. This indicates that the adsorption of tannic acid on silk is an exothermic process, and the affinity of tannic acid to silk decreases with increasing temperature, which is in agreement with the above-mentioned results in the investigation of the adsorption kinetics.

#### 3.1.4. Building-Up Property

The building-up property of bioactive compounds on textile fibers is related to their chemical structure, affinity and adsorption mechanism [3]. A good building-up property means a high utilization and a low production cost. The building-up property of tannic acid on silk fiber was evaluated by the exhaustion percentage and adsorption quantity. Figure 6 shows that the adsorption amount of tannic acid on silk fiber linearly increased with an increase in tannic acid concentration. It is worth noting that the exhaustion percentage and adsorption quantity of tannic acid reached to 57.14% and 142.84 mg/g, respectively, at the initial concentration of 25% owf, indicating the excellent building-up capability and high utilization of tannic acid.

### 3.2. Whiteness and Yellowness Indexes of the Tannic Acid Treated Silk

The whiteness index is an important indicator which affects the sensory evaluation of textiles. The white color is essential for casual silk apparel. However, the textile finishing processes often lead to the yellowing of the white silk fabric and cause the decrease of visual effect and commercial value. The small influence of chemical finishing on the color of white and light silk fabrics is an important concern. Thus, the effects of tannic acid on the whiteness and yellowness indexes of silk fabric were studied. Figure 7 shows that the treatment by tannic acid had a slight impact on the whiteness and yellowness indexes of silk fabric, especially when the concentration of tannic acid was lower than and equal to 5% owf. Compared with the untreated silk fabric, the sample treated with 5% owf tannic acid, which imparted good antibacterial and antioxidant properties to silk fabric (discussed later), almost had no change in whiteness and yellowness indexes.

### 3.3. Antioxidant Property of the Tannic Acid Treated Silk

Silk is usually in direct contact with skin when in use, so it is of great significance to develop healthy and hygienic silk textiles and materials with a good antioxidant function [18]. In our study, silk was endowed with antioxidant activity by the adsorption of tannic acid. As shown in Figure 8, the untreated silk fabric had a poor antioxidant activity of 32.3%. The silk fabric treated with 2% owf tannic acid exhibited an excellent antioxidant activity of over 99%. In other words, tannic acid on silk had the high efficiency in scavenging free radicals. The strong antioxidant activity of tannic acid is associated with the abundant hydroxyl groups in its structure [39]. After 10 and 20 washing cycles, the antioxidant activity of the silk fabric treated with 2% owf tannic acid decreased to 89.4% and 75.6%, respectively, due to the desorption of a small amount of tannic acid from silk in the washing solution. It is worth noting that the 5% owf tannic acid-treated silk had no obvious downward trend in antioxidant activity even after 20 washing cycles. The above experiments reveal that the antioxidant activity of the treated silk has relation to the quantity of tannic acid adsorbed by silk and the desorption of tannic acid from silk, and on the whole, the silk fabrics treated with 2% and 5% owf tannic acid have durable and good free radical scavenging capability.

### 3.4. Antibacterial Property of the Tannic Acid Treated Silk

Tannins have been proved to be effective in inhibiting the growth of microorganisms, which are dependent on the phenolic hydroxyl groups present in their molecules [26,29]. In the present work, the antibacterial activity of the tannic acid treated silk fabrics against *E. coli* and *S. aureus* was evaluated. Figure 9 shows that the untreated silk fabric exhibited poor antibacterial properties with the antibacterial activity of 29% and 25% against *E. coli* and *S. aureus*, respectively. The treated silk fabrics exhibited significantly enhanced antibacterial activity, which increased with an increase in initial tannic acid concentration. Notably, the silk fabric treated with 0.5% owf tannic acid had an excellent antibacterial property with the antibacterial activity of over 98% against *E. coli* and *S. aureus*. Figure 9 also shows the antibacterial durability of the 0.5% owf tannic acid treated silk subjected to 10 and 20 washing cycles. It can be seen that the change in the antibacterial activity of the treated silk induced by washing was not apparent, implying the excellent washing durability of antibacterial property.

In order to further demonstrate the excellent antibacterial ability of tannic acid, the relationships between the adsorption quantity of various natural compounds on silk and the antibacterial activity of the treated silk against *E. coli* were compared according to our previous reports [3,8,18]. As shown in Figure 10, compared with baicalin, quercetin, rutin, *Rheum emodi*, *Gardenia* yellow, and curcumin, tannic acid imparted the highest antibacterial activity to silk at the lowest adsorption amount. A similar result was found for the antibacterial property of the treated silk against *S. aureus*. Tannic acid contains a higher number of phenolic hydroxyl groups than other natural compounds, which possibly contributes to its excellent antibacterial ability.

## 4. Conclusions

This work provides a facile, efficient, and eco-friendly route to prepare bioactive and durable silk materials using tannic acid (a specific form of a hydrolysable tannin), and studies the adsorption properties and mechanism of tannic acid. The ion-ion interaction between tannic acid and silk contributed to the exothermal chemisorption of tannic acid on silk fiber. The adsorption of tannic acid on silk fiber was greatly dependent on pH and temperature, indicating that the control of these two parameters is of importance for the preparation of bioactive silk materials. Tannic acid exhibited an excellent building-up capability and a high utilization on silk during dip processing, and had less influence on the whiteness of silk fabric. The tannic acid-treated silk fabrics exhibited excellent and durable antibacterial and antioxidant activities. The high efficiency of tannic acid in inhibiting the growth of bacteria and capturing the free radicals enables the treated silk to become a promising healthy and hygienic material.

## Figures and Tables

**Figure 1 polymers-11-00970-f001:**
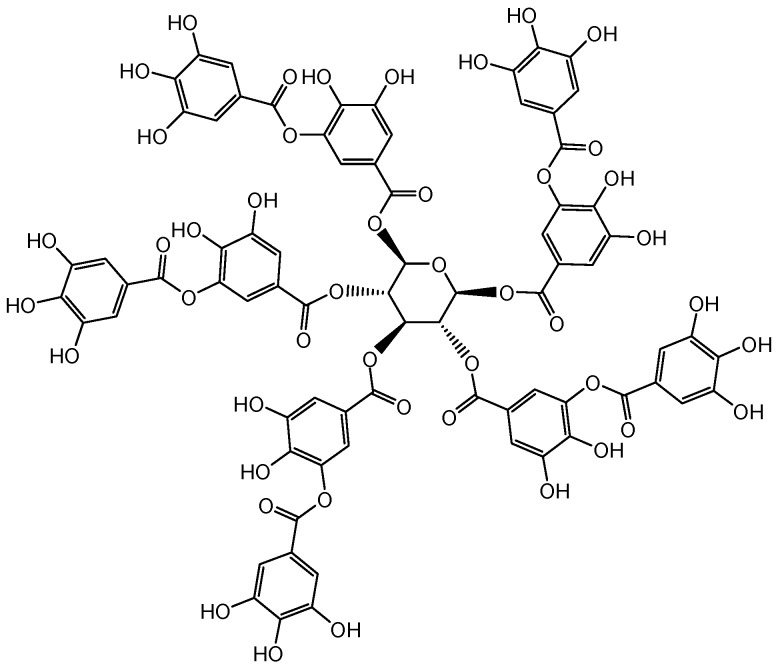
Chemical structure of tannic acid.

**Figure 2 polymers-11-00970-f002:**
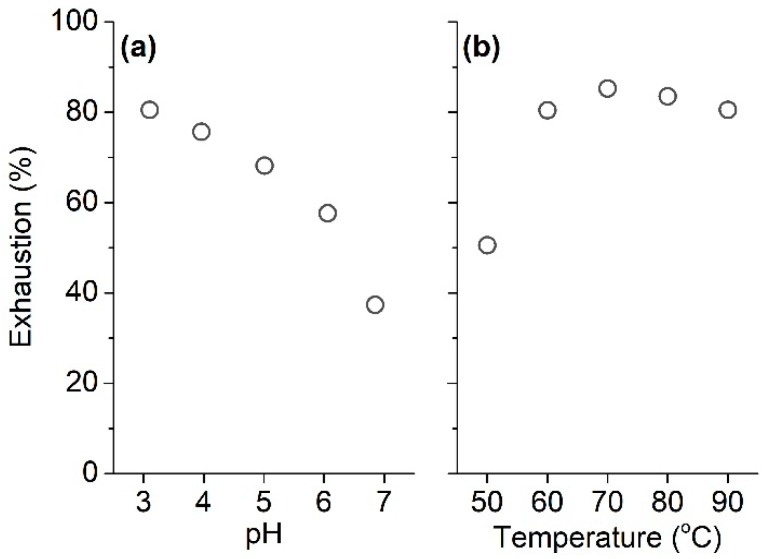
Effect of pH (**a**) and temperature (**b**) on the adsorption of tannic acid on silk.

**Figure 3 polymers-11-00970-f003:**
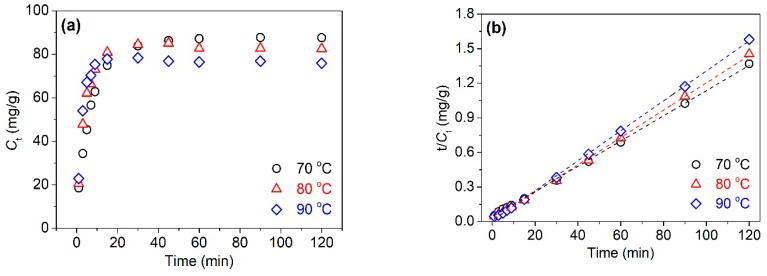
Adsorption rates of tannic acid (**a**) for silk and it plots as per the pseudo-second-order equation (**b**) at three temperatures.

**Figure 4 polymers-11-00970-f004:**
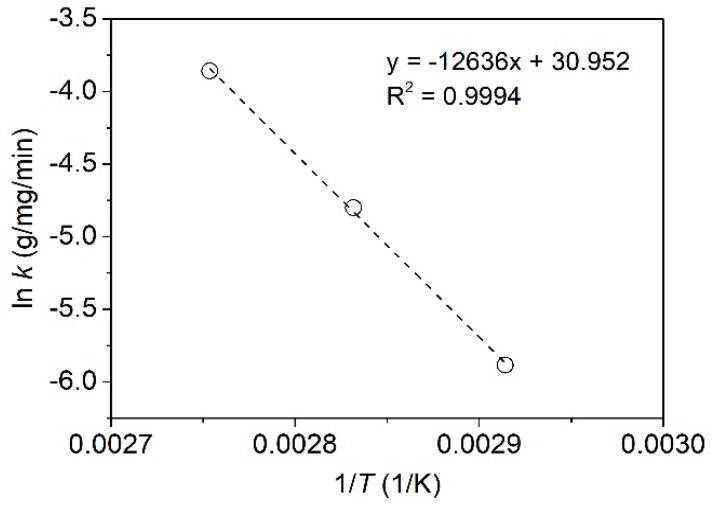
Plot of the Arrhenius equation at three temperatures.

**Figure 5 polymers-11-00970-f005:**
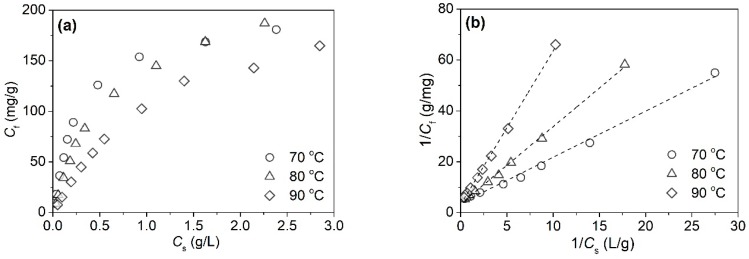
Adsorption isotherms (**a**) and Langmuir fitted curves (**b**) of tannic acid on silk fiber at three temperatures.

**Figure 6 polymers-11-00970-f006:**
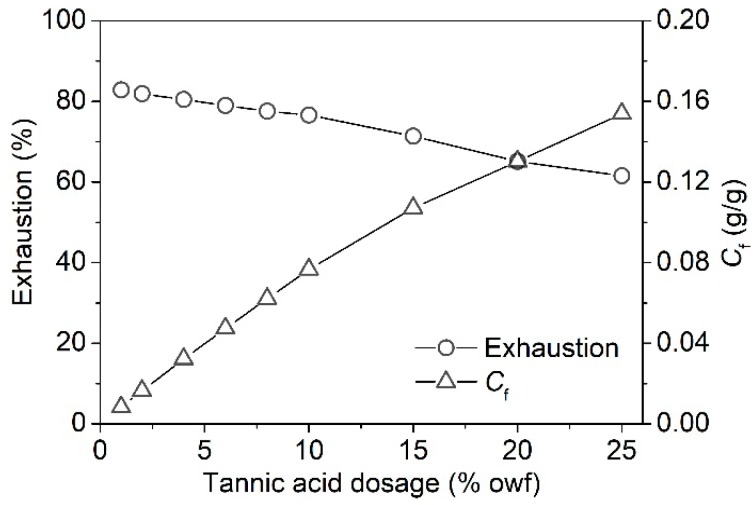
Building-up property of tannic acid on silk fiber.

**Figure 7 polymers-11-00970-f007:**
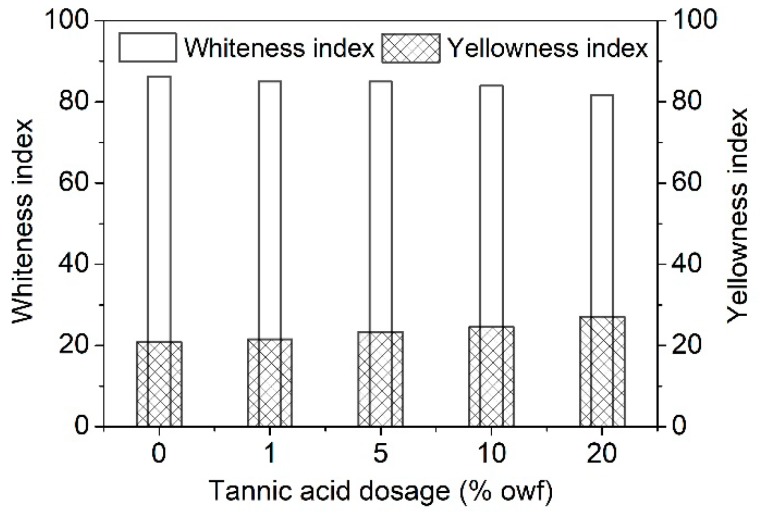
Whiteness and yellowness indexes of the silk fabrics treated with different concentrations of tannic acid.

**Figure 8 polymers-11-00970-f008:**
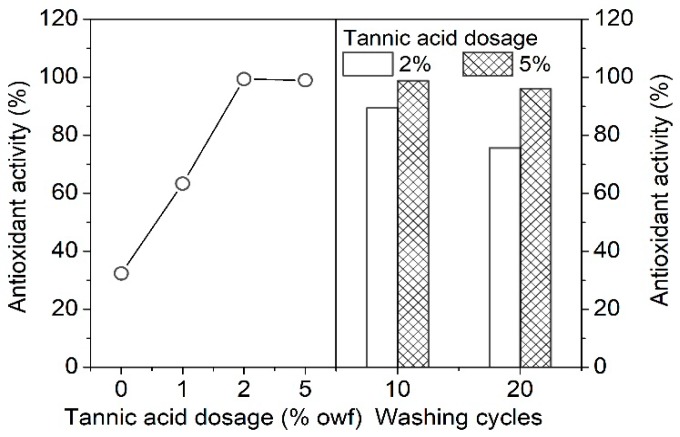
Antioxidant activity of the silk fabric treated with different concentrations of tannic acid, and the washing durability of the silk fabrics treated with 2 and 5% tannic acid.

**Figure 9 polymers-11-00970-f009:**
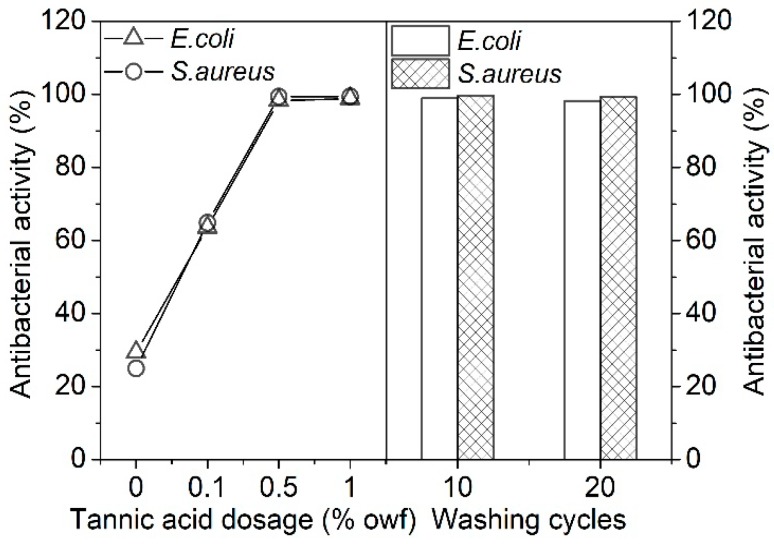
Antibacterial activity of the silk fabrics treated with different concentrations of tannic acid, and the washing durability of the silk fabric treated with 0.5% tannic acid.

**Figure 10 polymers-11-00970-f010:**
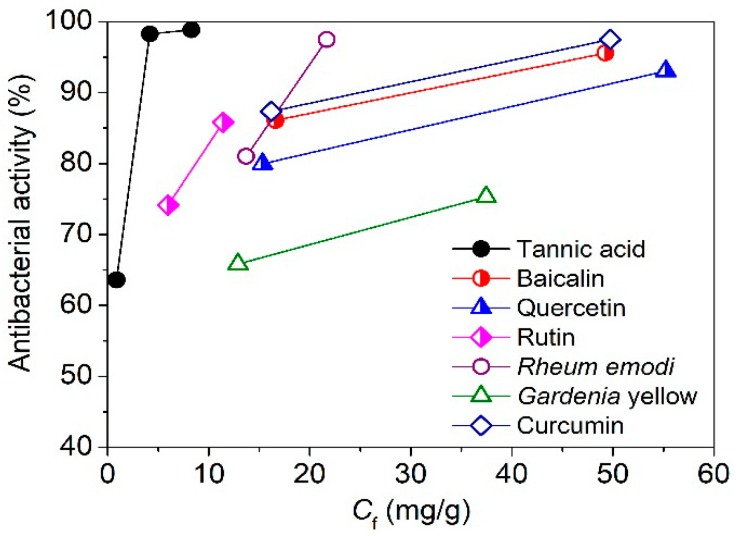
Correlation of the antibacterial activity of the treated silk against *E. coli* with the adsorption quantity of natural compounds on silk.

**Table 1 polymers-11-00970-t001:** Kinetic parameters of tannic acid obtained from the pseudo-second-order model.

Temperature (°C)	*r*_i_ (mg/(g·min))	*t*_1/2_(min)	*K* (g/(mg·min))	*C*_∞_ (mg/g)	*R*^2^
70	23.42	3.92	0.00278	91.74	0.9994
80	58.14	1.45	0.00823	84.03	0.9993
90	125.00	0.62	0.02113	76.92	0.9995

**Table 2 polymers-11-00970-t002:** Langmuir and Freundlich adsorption parameters of tannic acid on silk.

Temperature (°C)	Langumuir			Freundlich
	*K*_L_ (L/g)	*S* (mg/g)	*R*^2^	*R*^2^
70	0.84	0.55	0.9891	0.9054
80	0.64	0.46	0.9957	0.9558
90	0.45	0.36	0.9994	0.9738

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
