# Peer review of "Adsorption, Antibacterial and Antioxidant Properties of Tannic Acid on Silk Fiber"

_polymers, 2019, doi:10.3390/polym11060970_

Round 1
Reviewer 1 Report
Manuscript Title: Adsorption, antibacterial and antioxidant properties of tannic acid on silk fiber
The manuscript is good research work and can be accepted for publication readers after following major revision.
(1) Authors need to incorporate some recent reference to make it more interesting for readers.
(2) What is the significate of introducing antioxidant material in Textile silk?
(3) Author need to add graphic/pictorial or schematic presentation of the work to make it easily understandable.
(4) Author needs check manuscript thoroughly; specially for spelling mistakes and abbreviations (oC etc).
(5) Why author used only pH 3 to study the absorption kinetic.
(6) Author need to explain stability of Silk and tannic acid at different pH values and temperatures ((low to high)
(7) Author need to provide FT-IR for more conformation about absorption process.
(8) Why sorption capacity of tannic acid on silk depended appreciably on the pH of tannic acid solution? Author need to provide proper mechanism with reference.
(9) Author need to compare their results (absorption/antibacterial) with already reported similar research in tabulated for.
Author Response
Please read the attached files.

Reviewer 2 Report
I recommend this paper should be polished up focusing on:
(1) detailed explanation on comments,
(2) Objective conclusions based on results.
(3) detailed explanation of statistical analysis of results and is resubmitted
Authors have prepared bioactive and durable silk materials by the introduction of tannic acid using a facile adsorption process. The effects of pH and temperature on the uptake of tannic acid by silk were explored, the adsorption process and mechanism of tannic acid were studied using thermodynamic models to fit the adsorption data. The antibacterial and antioxidant activities as well as their washing durability of the treated silk fabric were analyzed also.
· The experimental design is not described well, especially in 2.2. Adsorption experiments relating to pH and temperature range. How did you determine experimental conditions?
· ABTS method doesn't have a reference method
· Antibacterial activity evaluation doesn't have a reference method. Just mentioned GB/T 20944.3–2008 without reference is not appropriate
· Why the incubator temperatures were 30 oC for E. coli solution and 24 oC for S.
aureus solution?
· What is the CFU used for bacteria? What nutrient agar was used?
· How were bacterial colonies growing on the petri dishes recorded?
· Antibacterial activity in text is refferes as reduction rate and showed in Figure 9., but it should be uniformly
· antimicrobial activities aren't described well
· There is no standard antibacterial compound as a positive control, only negative control wtih untreated fabrics
· Figures should be clearly presented
· The Antimicrobial activity results were not elucidated well in the discussion
· The sentence
on page 4 line 153 „This implies that tannic acid can be adsorbed by silk by virtue of the ion-ion interaction between the positively charged amino group in silk and the negatively charged hydroxyl group in tannic acid.“ Should be supported with arguments, especially because it was repeated in Conclusion section.
· Type of Statistical analysis isn't mentioned at all, either the Statistical program.
· In my humble opinion, this paper seems better for submission in Journal Applied Sciences
Author Response
Please read the attached files.

Round 2
Reviewer 1 Report
The manuscript "Adsorption, antibacterial and antioxidant properties of tannic acid on silk fiber"
looks good after revision and recommended for publication after following minor change.
1. Author need to correct degree centigrade (oC) throughout the manuscript.
2. Author need to include following References related to the subject.
International journal of biological macromolecules 87, 460-465
Chemical Communications 49 (78), 8818-8820
Author Response
Q1: Author need to correct degree centigrade (oC) throughout the manuscript.
A1: I guess, the reviewer thinks that the typewriting of degree centigrade is not correct. So, in the revised manuscript, the degree centigrade, oC was replaced by ℃. If there is any problem, I think the standard typewriting can be used during the production of the paper.
Q2: Author need to include following References related to the subject: International journal of biological macromolecules 87, 460-465; Chemical Communications 49 (78), 8818-8820
A2: In the revised manuscript, the mentioned references have been added in the introduction section by red marks.
Revised introduction section:
----- Bioactive products derived from natural resources, for instance flavonoids [3], chitosan [4], oxidized cellulose and carboxycellulose [5,6], and vitamins [7], are gaining great attention owing to their non-toxic, degradable, and eco-friendly performances. ------
[5] Sharma, P.R.; Varma, A.J. Functional nanoparticles obtained from cellulose: engineering the shape and size of 6-carboxycellulose. Chem. Commun. 2013, 49, 8818–8820, doi: 10.1039/c3cc44551h.
[6] Sharma, P.R.; Kamble, S.; Sarkar, D.; Anand, A.; Varma, A.J. Shape and size engineered cellulosic nanomaterials as broadspectrum anti-microbial compounds. Int. J. Biol. Macromol. 2016, 87, 460–465, doi:10.1016/j.ijbiomac.2016.02.024.
Reviewer 2 Report
Authors corrected manuscript according to comments.
Author Response
According to the other reviewer’s suggestion, two new references have been added in the introduction section, and the typewriting of degree centigrade has been improved in the revised manuscript.
Additionally, we have read and checked the manuscript again.